# Improving Dry Matter Intake Estimates Using Precision Body Weight on Cattle Grazed on Extensive Rangelands

**DOI:** 10.3390/ani13243844

**Published:** 2023-12-14

**Authors:** Hector Manuel Menendez, Jameson Robert Brennan, Krista Ann Ehlert, Ira Lloyd Parsons

**Affiliations:** 1Department of Animal Science, South Dakota State University, Rapid City, SD 57703, USA; hector.menendez@sdstate.edu (H.M.M.III); ira.parsons@sdstate.edu (I.L.P.); 2Department of Natural Resource Management, South Dakota State University, Rapid City, SD 57703, USA; krista.ehlert@sdstate.edu

**Keywords:** precision weighing, beef cattle, precision technology, precision system models

## Abstract

**Simple Summary:**

As precision livestock technology continues to become viable for extensive rangeland systems. it is important to determine which technology has the potential to positively impact grazing management. Utilizing precision weighing systems on rangeland beef cattle provides new and novel insight into individual animal performance throughout grazing periods, which are directly linked to stocking rates (cattle/ha/time), to ensure adequate forage dry matter intake for cattle while avoiding negative environmental impacts. Refining stocking rate estimates using precision body weight measurements and precision system modeling is critical as this management decision is fundamental to rangeland management and livestock productivity across the United States.

**Abstract:**

An essential component required for calculating stocking rates for livestock grazing extensive rangeland is dry matter intake (DMI). Animal unit months are used to simplify this calculation for rangeland systems to determine the rate of forage consumption and the cattle grazing duration. However, there is an opportunity to leverage precision technology deployed on rangeland systems to account for the individual animal variation of DMI and subsequent impacts on herd-level decisions regarding stocking rate. Therefore, the objectives of this study were, first, to build a precision system model (PSM) to predict total DMI (kg) and required pasture area (ha) using precision body weight (BW), and second, to evaluate differences in PSM-predicted stocking rates compared to the traditional herd-level method using initial or estimated mid-season BW. A deterministic model was constructed in both Vensim (version 10.1.2) and Program R (version 4.2.3) to incorporate individual precision BW data into a commonly used rangeland equation using %BW to estimate individual DMI, daily herd DMI, and area (ha) required to meet animal DMI requirements throughout specific grazing periods. Using the PSM, differences in outputs were evaluated using three scenarios: (1) initial BW (business as usual); (2) average mid-season BW; and (3) individual precision BW using data from two precision rangeland experiments conducted at the South Dakota State University Cottonwood Field Station. The data from the two experiments were used to develop PSM case studies. The trial data were collected using precision weight data (SmartScale™) collected from replacement heifers (Case study 1, *n* = 60) and steers (Case study 2, *n* = 254) grazing native rangeland. In Case study 1 (heifers), Scenario 1 versus Scenario 3 resulted in an additional 73.41 ha required. Results from Case study 2 indicated an average additional 4.4 ha required per pasture when comparing Scenario 3 versus Scenario 1. Sensitivity analyses resulted in a difference between maximum and minimum simulated values of 27,995 and 4265 kg forage consumed, and 122 and 8.9 pasture ha required for Case studies 1 and 2, respectively. Thus, results from the scenarios indicate an opportunity to identify both under- and over-stocking situations using precision DMI estimates, which helps to identify high-leverage precision tools that have practical applications for enhancing animal and plant productivity and environmental sustainability on extensive rangelands.

## 1. Introduction

Measuring the dry matter intake (DMI) of grazing beef cattle is a significant challenge from basic and applied research and production aspects. For estimating forage utilization and the amount of land area required for cattle grazing extensive rangelands, it is essential to determine the animal unit months (AUMs). For AUMs, the amount of forage needed per animal unit (AU) is defined as a 454 kg cow consuming 2.6% of BW on a dry matter basis, thus equating to 354 kg forage being consumed over one month (1 AUM) [1]. The DMI is required to determine stocking rates and supplementation requirements. The variation in estimated DMI of grazing cattle introduces error at the herd level since estimated DMI is based on full BW multiplied by a percentage of BW. The percentage of BW varies depending on animal class, production phase (e.g., lactating vs. dry), and forage quality [2]. For example, a lactating cow eating low-quality forage (<52% total digestible nutrients) consumes approximately 2.2% of BW on a dry matter basis, whereas a lactating cow eating high-quality forage (>59% total digestible nutrients) consumes 2.7% of BW on a dry matter basis [2,3]. Differences in DMI and BW can also significantly affect stocking rates. For example, a 227 kg heifer consuming 2.5% of BW on rangeland with 700 kg ha^−1^ available for consumption (i.e., 25% of total forage; 2800 kg ha^−1^) would require 0.24 ha AUM^−1^, whereas a 381 kg heifer consuming 2.5% of BW would require 0.42 ha AUM^−1^.

Traditionally, stocking rate estimates have been based on knowledge of rangeland stocking capacity from land manager experience, initial animal BW (if known), peak biomass (kg ha^−1^), and forage utilization measurements. Frequently, initial herd level average BW is used to calculate AUMs and, subsequently, stocking rate; however, herd level averages may not adequately account for the variability in initial individual animal BW, changes over time due to daily cattle growth (BW + Δ kg d^−1^ (daily gain)), differences in % BW of DMI, or changes in BW due to environmental conditions. Not accounting for this individual animal variability can have a proportional scaling effect from individual livestock operations to landscape scale estimates of AUMs required. The exclusion of individual animal BW data can result in potentially overgrazing forage resources and subsequently have negative impacts on natural resources and animal production.

Range and animal scientists have envisioned in-pasture weighing systems since the 1960s [4,5]. However, the advent of modern precision livestock technology has made in -pen and pasture weighing systems a viable option for research and production [6,7]. The advent of precision data collection for rangeland cattle, including the imminent development of camera-based weighing systems [7], has made it possible to weigh cattle on pasture in real time and provides new insight into individual animal weights throughout the grazing season [8,9]. These data provide the potential to match more closely the stocking rate and carrying capacity of rangeland for economically and environmentally sustainable grazing livestock production. As the role of precision technology grows in extensive rangeland systems, a critical question is how previously unattainable data can be leveraged in precision system models (PSM) [10]. A PSM model is specifically designed to incorporate precision livestock data to help inform management. Using PSMs will help evaluate complex tradeoffs relative to ranch management objectives such as animal efficiency, managing variability in forage resources (surplus and shortfall), environmental impact, and mental models. Deploying a PSM to estimate individual DMI using precision weighing technology may help to identify performance gaps in stocking rates, maximize forage utilization, and prevent overgrazing. Therefore, the objectives of this study were to (1) build a PSM to predict forage DMI and stocking rate in extensively managed cattle using precision BW; and (2) conduct two case studies to compare stocking rate predictions between PSM and traditional methodologies.

## 2. Materials and Methods

### 2.1. Study Site

The SDSU Institutional Animal Care and Use Committee approved all procedures involving animals (Approval #2109-054E and #2104-021E). Both case study projects were conducted at the South Dakota State University (SDSU) Cottonwood Field Station (CFS; Cottonwood, SD, USA, GIS cords: 43.989107 N, −101.857228 E), located in the Northern Great Plains and consisting of mixed grass prairie, where dominant forage species included western wheatgrass (*Pascopyrum smithii* (Rydb.) A. Love), crested wheatgrass (*Agropyron cristatum* (L.) Gaertn.), green needlegrass (*Nassella viridula* Trin.), and needle-and-thread (*Hesperostipa Comata* Trin. and Rupr), with the inclusion of sedges (*Carex* spp.), buffalograss (*Bouteloua dactyloides* (Nutt.) J.T.Columbus), and blue grama (*Bouteloua gracilis* (Willd. Ex Kunth.) Lag. Ex Griffiths) [11]. There are also recent introductions of non-native grasses, such as Kentucky bluegrass (*Poa pratensis* Boivin and Love) and Japanese brome (*Bromus japonicus* Thunb.). Elevation at the CFS ranges from 710–784 m, and the climate is an arid cold steppe [12]. The long-term average annual precipitation for the area is 468 mm (1992–2022) [13].

### 2.2. Precision System Model Development

Mathematical models are becoming increasingly critical for the application of precision livestock technology as they allow the users to ask “what if” questions, such as the question in the current study of how individual BW across grazing periods impacts DMI estimates [14,15]. As the granularity of data increases, it is essential that models are used not only to generate results but that the results are meaningful, providing insight that translates into management interventions [16]. A critical step in fully leveraging mathematical models is to start with a simple modeling approach which ensures that the model achieves its intended purpose. In the current study, we use a simple deterministic model (in terms of mathematical complexity and number of parameters and equations) to assess differences in DMI, grazing area required (ha), and forage consumed (kg), setting the foundation for more complex integration of grassland dynamics [17]. Thus, as the amount of livestock, forage, soil, and climate data (amongst many other sources) become available through precision technology, the most successful models will likely be the ones that are based on a clear understanding of the production system. This understanding will enable the development of meaningful and scalable decision support tools that are based on a clear (i.e., simple) foundation of science such as the basic principles of rangeland management for grazing livestock.

A PSM model evaluating forage consumption (AUM) was constructed in Vensim DSS™, a dynamic visually based modeling software, utilizing equations described in the online SDSU Grazing Calculator [3]. A “precision data” AUM model (AUM_PSM_) component was built in Vensim DSS to integrate daily individual BW data. The model was also re-constructed in Program R [18] to facilitate open-source PSM development and use. Model results matched both programs [Vensim DSS (version 10.1.2) and Program R (version 4.2.3)]. A 3D smoothing function was applied to raw weight data to minimize variation due to rumen-fill. The AUM_PSM_ included a discrete (daily time step, delta time = 1) first-order differential equation to aggregate the daily estimated DMI and hectares needed for each animal into monthly herd level values. Thus, the model output was total hectares needed per month to meet cattle nutrient requirements and total forage consumed at a herd level. Fixed parameters for the model can be found in Table 1 and Table 2. Animal BW data were used from two grazing studies. Case Study 1 utilized data collected from a replacement heifer development study managed on dormant forage. Case Study 2 utilized data collected from summer grazed yearling steers.

### 2.3. Case Study 1

Replacement Angus heifers (*n* = 60; initial BW = 237.6 ± 15.5 kg) grazed dormant native pastures (*n* = 2; 115.1 ha and 93.4 ha) from November 2021 to May 2022, as part of a broader project to integrate precision feeding technology to precisely manage heifer development [19] (Figure 1). Individual animals were tagged with a radio frequency identification device (RFID; Allflex Inc., Dallas, TX, USA) to pair weight measurements to each animal. Both treatment groups were supplemented with 2.27 kg hd^−1^ d^−1^ of pelleted dried distiller’s grains with solubles (DDGs). The supplement was delivered to the control group via a traditional bunk fed method, and the precision group supplement was offered via a Super Smartfeed Producer^TM^ (C-Lock Inc., Rapid City, SD, USA). Individual BWs were calculated daily from partial BWs measured using a front-end scale (Smartscale™, C-Lock Inc. Rapid City, SD, USA; Figure 1) positioned at the water source. Forage samples were collected at the beginning and end of the trial using a grid sampling technique for each pasture (*n* = 10 per pasture; 0.25 m^2^ quadrat) to provide initial available forage values (kg dry matter ha^−1^). In Case study 1, only the initial forage value was used to estimate stocking rate.

### 2.4. Case Study 2

Steers were managed on native pastures as part of a long-term (>80 year) grazing study evaluating the effects of stocking density on plant communities and animal performance. The current project’s broader objectives are to evaluate opportunities for precision livestock technology to improve animal performance, monitor energy expenditure, and mitigate environmental impacts in extensively grazed cattle. More detailed methods are described by Vandermark [20]; briefly, crossbred Angus steers (*n* = 254) were managed on native pastures from June to August of 2021 and 2022, respectively. Each steer was weighed, fitted with an RFID tag, and allocated to one of six pastures equipped with a precision scale system (SmartScale™, C-Lock Inc., Rapid City, SD USA) positioned at the water source. Each pasture was assigned either a rotational (RG) or continuous (CG) grazing strategy and one of three stocking rates—low (pasture area = ~70.31 ha), medium (pasture area = ~53.15 ha), and high (pasture area = ~30.77 ha); 0.3, 0.42, and 0.7 AUM’s, respectively—in a 2 × 3 factorial design. The pasture sizes and stocking rates (above) were not used in the PSM_AUM_ simulations; rather, only the precision BW and initial forage values were utilized. The purpose of describing the differences in stocking rates is to emphasize that we utilized diverse sources of BW data compared to only a single stocking rate, as different stocking rates have been shown to impact rates of BW gain [11]. Forage samples were collected using random clippings (*n* = 5, 0.25 m^2^ quadrat) in each pasture at bi-weekly intervals for both grazing trials (2021 and 2022) to determine peak biomass.

### 2.5. Precision System Model Application

Individual animal BW data for both case studies were downloaded through an Automated Programming Interface (API, C-Lock Inc.) into Program R [18]. Weight data were assigned to individual animal records using the data.table package [21], filtered for spurious weight data points utilizing robust regression [22], and organized into longitudinal data frames. Static values (i.e., the same BW for each animal each day) were used to compare the AUM model outputs and the SDSU Grazing Calculator outputs to ensure mathematical accuracy and that double-accounting was avoided. It is important to note that the size or nutritional value of individual pastures for Case study 1 (*n* = 2) and Case study 2 (*n* = 6 per year) were not used in the PSM_AUM_; rather, only the initial (Case study 1) or estimated peak standing forage (Case study 2) and individual precision BW were used to estimate total grazing area needed and the total forage consumed for each of the three scenarios. Using initial dormant forage or estimated peak forage production are commonly employed by rangeland managers for setting stocking rates before turning livestock out to graze.

Three scenarios (described below) were simulated to determine differences in calculated total hectares of pasture needed over each grazing period. The first two scenarios were designed to represent typical calculations used to determine stocking rate, while the third utilized precision BW data. All simulations were based on the DMI estimation method using %BW described above. Model parameters were the same in each scenario (Table 1). Scenario 1 applied traditional rangeland AUM estimation methods based on average initial BW (Table 3) at the beginning of each grazing period. Scenario 2 consisted of calculating a mid-season average BW for the heifers and the steers. Case study 1 was calculated using the desired BW weight at time of breeding in May of 381 kg minus the initial average herd BW (Table 3). Unlike the heifers, the steers in Case study 2 did not have a specific desired end BW, as one of the full experiment’s goals was to assess differences in BW across stocking rates, not necessarily to maximize performance [20]. Thus, the steer mid-season BW was calculated using an average of the average initial herd BW (day 1) and ending herd BW (day 60) (Table 3). The impetus behind this scenario was to account for changing BW at a herd level at mid-season in an attempt to capture the same expected variation in BW throughout the grazing period. Finally, Scenario 3 utilized daily BW measurements from the precision scales for each animal over the total length of the grazing period (Table 3) with the AUM_PSM_.

Finally, we conducted a Monte Carlo analysis (10,000 runs) to determine the percentiles of potential variation from %BW on AUM_PSM_ calculations on a daily basis impacting total hectares required and total forage used for each grazing period. The BW parameters ranged from 1.8 to 2.7% using a univariate normal distribution.

## 3. Results

### 3.1. Case Study 1

The initial BW of replacement heifers was 243.45 kg, with a mid-season BW of 301.53 kg per head. It had a 175 d grazing period; 8 d were removed due to missing BW due to system malfunction, resulting in a 167 d period used for this analysis. The greatest difference in total area needed was 73.41 ha between Scenarios 1 and 3; however, using an estimated mid-season average (Scenario 2) only resulted in 10.03 ha less than the precision estimate (Scenario 3; Table 4; Figure 2). Simulations were deterministic and did not have a probability distribution. The sensitivity analysis results of forage consumption from variation in %BW ranged from 56,019 to 84,014 kg on day 167, increasing as the grazing trial progressed (Figure 3). Like estimated forage consumption, the sensitivity analysis of total hectares required, from variation in %BW, ranged from 244 to 366 ha on day 167 and increased as the trial progressed (Figure 3).

### 3.2. Case Study 2

Average initial and mid-season BW are reported in Table 3, with an average initial BW of 293 ± 49.6 and 369 ± 37.1 kg across six pastures for the year 2021 and 2022, respectively. Study length was 68 and 76 d for trial year 1 and 2, respectively; however, due to equipment failures and poor visitation rates, 8 and 16 d were removed from the end of each trial period to obtain a 60 d period for the analysis of each. The average difference was 4.4 ha per pasture between Scenarios 3 and 1; however, using estimated mid-season BW only resulted in a 0.9 ha difference between Scenarios 3 and 2 (Figure 4; Table 4). Sensitivity of average DMI due to variation in %BW ranged from 8530 to 12,795 and 7182 to 10,774 kg per pasture on trial day 60 for steer trials 1 and 2, respectively. This directly resulted in the range of total required ha per pasture of 17.6 to 26.3 and 17.7 to 26.6 ha on day 60 (Figure 5) for steer trials 1 and 2, respectively.

### 3.3. Considerations for Range Beef Cattle

As expected, considerable differences existed between the traditional AUM method and the precision-informed AUM_PSM_ method, caused by individual animal BW. Differences between Scenarios 1 and 3 were amplified over time as BW became increasingly influential to AUM estimates. This indicates that obtaining initial BW before turn-out for grazing is a critical management factor and that precision weighing can aid both traditional and precision AUM estimates [23]. A mid-season average (Scenario 2) captured similar results as the precision-informed AUM estimate (e.g., ~3% difference for heifers). However, this agreement was found using expected daily gain based on a well-developed nutrition plan (grazing and supplement) and from a sample population that was culled to represent a similar weight range. It is unlikely that such cattle uniformity exists for the typical rancher; therefore, we would expect more variability. Thus, an opportunity exists to evaluate how many ranchers use initial or mid-season BW and the different qualities of BW data available. It is essential to evaluate the usefulness of tools like precision weighing for effective adoption and avoidance of technologies that do not yield returns.

Precision system modeling focuses on identifying high-leverage precision measurement or management tools to minimize a performance gap [5]. Grazing models like APEX use growth functions based on mean animal weights [24]. Although models, like APEX, can incorporate probabilistic functions, the variation applied is random for each animal at any given time-point, which does not reflect grazing behavior or environmental responses affecting performance, while precision BW data is representative of such variations. Thus, the next step in PSM is to identify feedback mechanisms that more adequately represent individual BW variation. These mechanisms will likely include other data streams (e.g., climate data effects on heat/cold stress) to model this individual variation. Thus, PSM will help to further separate rangeland animals into different groups of efficiency/quality. For example, the variation of total forage consumed, and hectares needed caused by the sensitivity analysis of %BW for each heifer or steer (Figure 3 and Figure 5) indicates that individual animal consumption rates (kg dry matter d^−1^) are impactful. As more information is collected about feed-efficient grazing cattle, there is the potential to select animals with similar efficiency levels to minimize this variation.

Applying precision data and derived model coefficients provides more precise DMI estimates and shows the advantages and disadvantages of herd level estimates, especially in the context of changing cattle demands and forage nutrient composition and availability throughout the grazing season. For instance, unlike the data used in the current study, where forage availability was a fixed resource (i.e., dormant forage), most livestock grazing capitalizes on seasonal changes in forage availability and quality [25]. Thus, the next step is to account for seasonal changes of biomass availability and nutrient composition of the heifer and steer case studies instead of only using initial dormant forage availability or peak biomass estimates. Evaluating precision informed stocking rates for continuous versus rotational grazing is another opportunity for further research using a more complex version of the PSM_AUM_. Precision BW estimates can help producers fine-tune stocking rates for future pastures (dynamic pasture adjustment using AUM_PSM_). Further, virtual fencing has made these potential AUM_PSM_ informed rotations feasible because physical labor and infrastructure (e.g., water) requirements are minimized [5].

Grazing livestock research and production are focused on the “margins” like performance and costs, with the aim to maximize rangeland resources [11]. It has been estimated that 20–40% of US grasslands are overgrazed [26]. Heavy grazing has been shown to be equally as profitable as more lightly grazed systems in years with average precipitation; however, limitations in forage productivity and water infiltration are extreme during drought within this type of grazing management system. Precision system models help to answer complex questions regarding the impact of precision livestock technologies on marginal increases in animal productivity at a local and supply chain level [27]. While a comparison of Scenarios 1 and 3 indicates local ranch-level benefits, especially for extensive systems (Figure 2 and Figure 4), viewing supply chain impacts supports efforts for marginal improvements at the local level (e.g., climate-smart commodities). For example, the current number of heifers in South Dakota (US) is 375,000 [28]. We extrapolated our results to a state level using the following Equation (1):(1)$$Hectares\;Overgrazed=\frac{{Heifer}_{s}}{{Heifer}_{l}}*Difference\;in\;Hectares,$$
where hectares overgrazed is the number of additional hectares required for a specific grazing period, *Heifers_s_* is the current number of heifers in the state, *Heifers_l_* is the number of Angus heifers used in the current study (*n* = 60), and the difference in hectares (Δ ha) is the difference between the modeled results of Scenarios 1 and 3 (Table 5).

It is important to note that the estimates provided in Table 5 are based on a specific grazing management system of “take-half-leave-half”, which is designed to avoid overgrazing and promote plant regrowth. The current modeled example (Equation (1); Table 5) is based off static values and does not account for potential forage regrowth after dormancy during the spring. Therefore, the overgrazing of 458,812 acres is relative to the “take-half-leave-half” grazing strategy and should not be applied as an indication of potential rangeland degradation from overgrazing for other production settings without accounting for grazing management and potential plant regrowth details. Rather, this estimate of overgrazed areas is only to highlight potential differences that may occur at scale, and which may potentially be improved through the use of precision-derived stocking rate coefficients for different classes of grazing beef livestock.

## 4. Conclusions

Using technologies like SmartScales™ to provide precision informed stocking rate coefficients is likely to provide a high-leverage management opportunity for rangeland managers. As data increase for animal classes and grazing periods (e.g., winter, summer), more precise stocking estimates can be obtained to maximize pasture use relative to production goals (harvest efficiency) [29]. However, research derived precision AUM coefficients may be sufficient for ranch managers as the implementation of scale-based precision technology is limited by cost and infrastructure within extensive systems (i.e., precision weighing is a means to an end). More investigation is needed for non-growing animal classes such as mature cows that are likely to have much less variation in DMI over the grazing season. Practical steps are needed to evaluate how precision data can be integrated into PSM to guide precision data collection efforts [30]. The next step is using precision-informed forage nutrient composition data from forecast or remote sensing (e.g., near real-time forage production and nutrient composition estimates) and DMI equations that include net energy for maintenance [2]. For example, in some years, grass may come out of dormancy before May, depending on the climate, plant root storage, and soil moisture conditions, which could alter available biomass and its nutrient composition [31] (e.g., increased protein and lower fiber, which was not the case for our heifer case study). Additionally, using PSM to evaluate local-ranch- and supply-chain-level impacts will provide quantitative justification for specific precision livestock tool use or future development.

## Figures and Tables

**Figure 1 animals-13-03844-f001:**
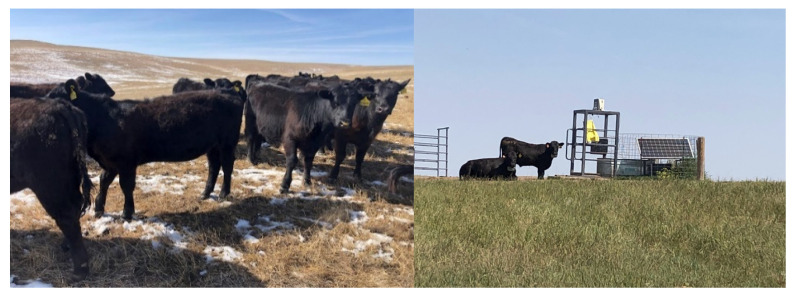
Dormant winter pasture and heifers (**left**) and steers using the SmartScale™ (**right**).

**Figure 2 animals-13-03844-f002:**
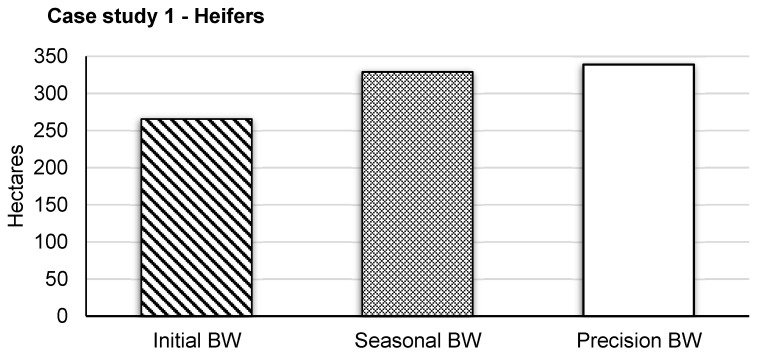
Simulated area (ha) to graze 60 heifers for 167 d based on body weight (BW) dry matter intake estimation in Scenario 1 (initial herd BW), Scenario 2 (herd mid-season BW), and Scenario 3 (precision BW).

**Figure 3 animals-13-03844-f003:**
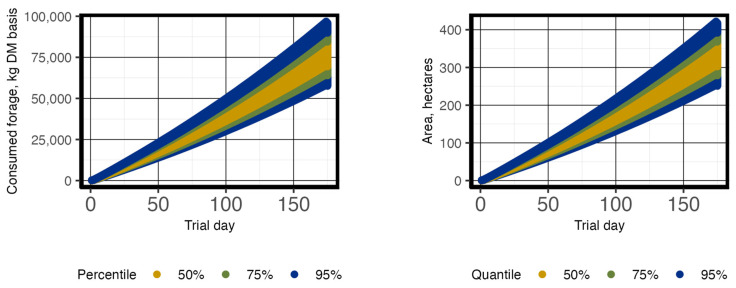
The percentiles of potential variation of daily total forage consumed (kg DM basis) from changes to % body weight (BW) (**left** panel) and required pasture size (**right** panel) over a 167 d period for replacement heifers. Initial animal unit equivalents were 0.54 based on average herd BW.

**Figure 4 animals-13-03844-f004:**
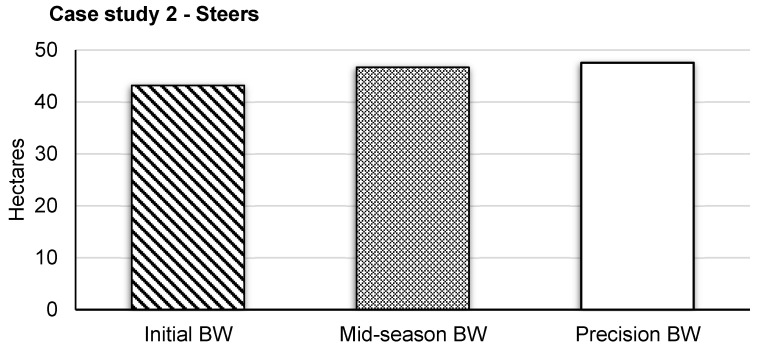
Simulated area (ha) to graze steers (average of 2021 and 2022 BWs; *n* = 254) in a typical pasture containing ~20 steers) for 60 d based on body weight (BW) dry matter intake estimation in Scenario 1 (initial herd BW), Scenario 2 (herd mid-season BW), and Scenario 3 (precision BW). Overall required grazing ha were calculated for each individual animal (~20) within each group (*n* = 6) per year (2021, 2022), and all groups (*n* = 12, i.e., six per year) were averaged to represent the total variation captured by individual steers within both trials (*n* = 254).

**Figure 5 animals-13-03844-f005:**
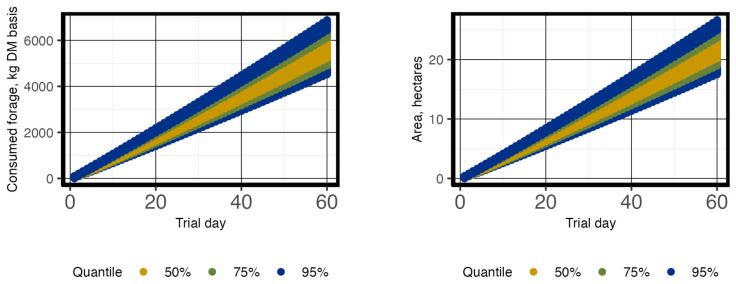
The percentiles of potential variation of daily total forage consumed (kg DM basis) from changes to % body weight (BW) (left panel) and required pasture size (right panel) over a 60 d period for steers. The sensitivity analysis represents the total number of steers (*n* = 254), similar to Figure 4 above. Initial animal unit equivalents ranged from 0.59 to 0.78 based on average herd initial BW of each pasture group (*n* = 6) of steers per each year (2021 and 2022; *n* = 2).

**Table 1 animals-13-03844-t001:** Model parameters used in the three scenarios for the heifer and steer stocking rate calculations based on the “take-half-leave-half” method suggested by the USDA Natural Resources Conservation Service (NRCS). The “take-half” consists of 50% of forage being taken (25% is consumed; 25% is trampled, urinated, and defecated on), while the “leave-half” leaves 50% of total biomass behind to regenerate plant growth.

Model Parameters	Heifer	Steer 1	Steer 2	Unit
Harvest Efficiency	0.25	0.25	0.25	Dimensionless
Days per Month	30	30	30	Day
Number of Cattle	60	127	135	Head
Percent BW	2.5	2.5	2.5	%

**Table 2 animals-13-03844-t002:** Average initial forage biomass of winter heifer pasture and estimated peak forage biomass within each summer steer pasture (kg per ha^−1^).

Case Study	Pasture	2021	2022
Case Study 1:			
Replacement heifers	–	–	917.78
Case Study 2:			
Steers	1	971.78	712.86
	2	935.91	1032.30
	3	1113.00	1305.79
	4	958.33	631.04
	5	1568.07	1064.81
	6	1713.78	1217.24

**Table 3 animals-13-03844-t003:** Scenario parameters for body weight (BW).

		Scenario					
Study	Pasture	1: Average Initial BW, kg hd^−1^		2: Mid-Season BW, kg hd^−1^		3: Individual Precision BW, kg hd^−1^ d^−1^	
Case Study 1:							
Replacement heifers	–	243.45		301.53		Individual weights	
Case Study 2:							
Steers		2021	2022	2021	2022	2021	2022
	1	352.33	273.66	375.71	300.89	–	–
	2	338.92	274.82	360.82	299.28	–	–
	3	345.80	271.23	366.92	302.58	–	–
	4	348.89	270.50	363.75	298.65	–	–
	5	352.91	271.68	378.12	300.47	–	–
	6	341.20	271.54	367.06	298.41	–	–

**Table 4 animals-13-03844-t004:** Total hectares required for each case study (replacement heifers and grazing steers) estimated according to total forage production (initial or peak), and intake estimated as a function of BW measured according to each of the three scenarios, initial BW, estimated mid-season BW, and individual precision BW.

		Scenario					
Study	Pasture	1: Average Initial BW, Hectares		2: Mid-Season BW, Hectares		3: Individual Precision BW, Hectares	
Case Study 1:							
Replacement heifers	–	265.77		329.17		334.07	
Case Study 2:							
Steers		2021	2022	2021	2022	2021	2022
	1	52.21	44.60	50.85	49.04	50.85	44.43
	2	52.73	48.78	56.14	48.78	52.89	41.74
	3	53.80	44.21	57.08	49.32	54.60	54.80
	4	51.70	42.08	53.89	48.97	48.55	42.02
	5	54.90	44.28	58.83	48.97	46.58	34.91
	6	55.61	46.27	59.83	50.85	56.33	42.96

**Table 5 animals-13-03844-t005:** Extrapolation of Scenarios 1 and 3 to estimate hectares required for a 167 d period at a state level based on January 2023 state heifer numbers.

Scenario	ha	Δ ha	State ha Required	Area Overgrazed ha
1	265.55	-	1,659,687	-
3	338.96	73.41	2,118,500	458,812

## Data Availability

The data presented in this study are available on request from the corresponding author.

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
