# Peer review of "Improving Dry Matter Intake Estimates Using Precision Body Weight on Cattle Grazed on Extensive Rangelands"

_animals, 2023, doi:10.3390/ani13243844_

Round 1

Reviewer 1 Report

Comments and Suggestions for Authors

The article “Improving Dry Matter Intake Estimates Using Precision Body Weight on Cattle Grazed on Extensive Rangelands” details and compares commonly applied methods for determining stocking rates on rangelands to a novel precision based BW method. In general, the paper was well written and brings to light the potential problems associated with basing stocking rates off of the initial body weight of growing animals. There is necessity for works such as this to help push technology application in production settings, or to at least provide better estimates when applying rules of thumb. Overall, it is a good paper but there are areas that need to be expanded upon with minor editing throughout.

L9 – suggest “which technologies have”. If using singular perspective suggest “which technology has”

L32 – change “steers” to “case study 2” for consistency.

L40 – Remove the definition of DMI. Suggest “(DMI) of grazing beef…”

L54-57 – This needs to be revisited, as the weanling stocker daily requirement is incorrect by my calculation (227 kg * 0.025*30.5)/700 kg ha-1 would be 0.25 ha AUM-1.

L66-68 – I also suggest mentioning how better data better enables matching of the stocking rate and grazing capacity of a rangeland, promoting environmentally and economically sustainable production.

L98 – Typically provide the most recent 20 – 30 years. I have no problem if the average has remained steady, just noticed this.

L127 – There is a problem with your citation.

L128 – Allflex*

L123-137 – It states that initial and final forage samples, but table 2 only provides a single initial value.  Do you think this accurately reflects the forage available through dormant and growing periods?

– The grazing period for CS1 was November – June, so what does the 2022 value represent, dormant winter forage, growing forage, or both? How was the stocking rate for CS1 calculated and was it adjusted at all during the grazing period?

–Wouldn’t it be more accurate to split into dormant and growing periods since they are innately different and intake of each as %BW would vary quite a bit?

– Were the heifers only on one pasture? How many grazeable acres did they have access to?

L148-162 – Similar to above, what are the pasture sizes and grazeable acres for each set of animals (at least an average size/area)?

– How were applied stocking rates calculated and were they held constant or adjusted for lower forage production in 2022?

– Was intake estimated indirectly from clippings in the work by Vandermark? If so, was that data applied here?

– Any auxiliary data such as forage allowance, forage accumulation rate, total forage production, etc. would be very beneficial.

L179 – There is a problem with your citation.

L181-182 – Wouldn’t mid-season average BW be the average weight of animals on the day that is the mid-point of the grazing period (e.g., grazing period of 60-d, mid-point = ~ 30 days)? From my perspective taking the average of the total gain is biased to those periods of time with greater ADG. If an overall average BW for the grazing period is what you are going for, I suggest calling it average BW instead of mid-season average BW.

L191 – Why was intake limited to 2.7%? Stocker intakes can be observed in the 3% range on high-quality forages. I have no real problem with this range, just curious as to why it was used.

L221 – Forgive my ignorance, but wouldn’t the simulated area required to graze have a range based upon estimated DMI like what is indicated in Figure 3? If so, could this not be indicated with “error” bars?

L237 – Are these figures not the same data but represented in different units? Could you not merge them by using two Y-axes?

L294 – How many animal units do these figures represent? Same comment about two Y-axes as above.

L331-332 – This needs to also be stated in the methods.

L358 – suggest using subscripts for Heiferss and HeifersL as it is difficult to read in current form.

L362 – I think this it is a large assumption to state ~half a million ha are being overgrazed solely based on AUM calculations. As you stated, this current simulation is not accounting for dynamic factors such as forage nutritive value or constituents that drive intake, nor the production stage of the animal. On a yearlong basis, across forage maturities and animal production stages, I would estimate the average intake of replacement heifers to be closer to the 2.4% range. This means that most stocking rates using AUE intake estimates (2.6% BW) would be more conservative than what is being applied. Also, this very much depends on what your definition of overgrazed is. As the paper assumes 25% utilization and waste rates (conservative rates) and 50% residual for plant growth. However, 50% is the point at which regeneration slows due to root shedding and regeneration processes, not that it is detrimental to the plant (at least major forage species). I like the table for illustrative purposes but feel this could possibly be misconstrued by certain groups.

Author Response

Thank you for the review. 

Reviewer 2 Report

Comments and Suggestions for Authors

The authors have addressed, to some extent, a pertinent and timely research question, showcasing a commendable level of expertise in the field. The manuscript is well-structured, with a clear and logical flow that aids in the reader's understanding. However, the methodology is partially robust, and the data analysis is supported only by deterministic models, lacking adequate scientific demonstration with replications on other types of pasture and animals. As a result, the contribution to scientific advancement in the field of precision grazing is modest. Therefore, I am compelled to provide a negative recommendation for publication as a research article. However, considering the growing interest in the field of precision livestock farming and the fact that the data and the type of modelling are still noteworthy, I suggest resubmitting it as a brief report.

I would like to highlight the absence of several citations in the text, replaced instead by the phrase "(Error! Reference source not found)." This issue is likely due to errors in the reference management software utilized. For this reason, I am unable to assess the adequacy of the bibliographic references that underpin the proposed research.

Author Response

Thank you for the review. 

Reviewer 3 Report

Comments and Suggestions for Authors

Abstract: Lines 16-23 can be minimized. The abstract is confusing when presenting the results and need to be improved. The main objective is to develop a model, but no results are provided as to what the model entails or how it's being used. Was there any analysis done regarding the significance of model parameters?

Line 25: how do you predict pasture? Are you predicting pasture use? Needs clarified. 

Line 30: which treatment was 73.41 ha referring to?

Line 56-57: are your AUMs correct? A heavier heifer usually requires more than a lighter one.

Line 69: GrowSafe has had a pasture weight system for years, not just recently

Line 85: materials and methods for case studies are extremely vague and do not provide sufficient information regarding the models and how they should be interpreted

Line 124: pastures aren't dormant in June, that's peak growing season. Need more information in the manuscript regarding pasture quality and production. 

Line 127: several references not found throughout, please remedy

Line 159: were the differences in stocking rates accounted for in the model. If not, data is confounded by treatment

Table 3: need to provide average weights from precision technology

Results: need to provide information regarding supplement intake, did each animal meet target intake, was this available throughout the entire grazing period, etc. Due to drastically different grazing conditions throughout the studies, suggest breaking down the PSM into monthly models and not entire trial periods.

Overall and interesting study, but is lacking in details. 

Author Response

Thank you for the review.

Round 2

Reviewer 2 Report

Comments and Suggestions for Authors

Dear authors,

I greatly appreciate the improvements you have made to the text. I am pleased to note a significant enhancement in the description of the methods and results, which now appear more clearly aligned with a robust statistical approach.

I was able to assess the bibliography following the correction of formatting errors, and I can provide a positive evaluation.

In light of these considerations and after a thorough re-evaluation of the submitted paper, contrary to my initial assessment during the first review, I can now provide a positive judgment.

Having no further concerns, I believe that the manuscript can be accepted for publication.